# Institutionalising urogenital schistosomiasis surveillance: Best practices to improve female genital and urinary schistosomiasis control in South Africa

Takalani Girly Nemungadi[1,2*], Tsakani Furumele[3], Absalom Mwazha[1,4], Myra Taylor[1], Saloshni Naidoo[1], Eyrun F. Kjetland[1,5]

**1** Discipline of Public Health Medicine, School of Nursing and Public Health, College of Health Sciences, University of KwaZulu-Natal, Durban, South Africa, **2** World Health Organisation, Emergency Preparedness and Response, Pretoria, South Africa, **3** Communicable Disease Control Directorate, National Department of Health, Pretoria, South Africa, **4** Department of Anatomical Pathology, National Health Laboratory Services, Durban, South Africa, **5** Department of Global Health and Department of Infectious Diseases Ullevaal, Oslo University Hospital, Oslo, Norway

\* takalaninemungadi@gmail.com

## Abstract

### Background

In the absence of an active schistosomiasis control programme, the affected community is vulnerable to complications such as female genital schistosomiasis. Research has shown that female genital schistosomiasis is a challenge faced by many African women including those from South Africa. Since 2008, the South African National Department of Health has been trying to resuscitate the schistosomiasis control programme; the programme has not been fully established or implemented. However, there are some surveillance best practices that the country can institutionalise to improve control.

### Materials and methods

A descriptive analysis of urogenital schistosomiasis data from the National Health Laboratory Services, Notifiable Medical Conditions Surveillance System, and District Health Information System was conducted in 2023. A document review was also carried out in 2023 to determine surveillance best practices to guide the establishment of sentinel sites for improving schistosomiasis and female genital schistosomiasis control.

### Results

The Health Laboratory Services, Notifiable Medical Conditions Surveillance System, and District Health Information System are the existing surveillance and reporting

**Data availability statement:** All relevant data are within the paper and its Supporting Information files.

**Funding:** This work was supported by the University of KwaZulu-Natal College of Health Sciences PhD Scholarship (student number 216073797) to TGN. The funders had no role in study design, data collection and analysis, decision to publish, or preparation of the manuscript.

**Competing interests:** The authors have declared that no competing interests exist.

systems. According to the Notifiable Medical Conditions Surveillance System (the overall and central notification system for the notifiable medical conditions), a total of 56529 urogenital schistosomiasis cases were reported nationwide between 2017 and 2021 (ranging from annual cases of 4140–15032). Most cases (>90%) were reported from public health facilities. The country's Regulations on the surveillance and control of notifiable medical conditions stipulate that schistosomiasis is one of the priority conditions that should be notified (within 7 days of clinical or laboratory diagnosis) by all public and private health care providers, as well as public and private health laboratories. The Regulations did not specify female genital schistosomiasis as one of the notifiable medical conditions. As a result, there was no reported data on female genital schistosomiasis and true burden was not known.

## Conclusion

The data collected through the National Health Laboratory Services, Notifiable Medical Conditions Surveillance System, and District Health Information System demonstrate that there are formalised schistosomiasis reporting systems, but no female genital schistosomiasis reporting. The existence and use of these surveillance systems demonstrate the country's potential to integrate the systems to enhance the prevention, surveillance, reporting, and management of schistosomiasis and introduction of surveillance for female genital schistosomiasis surveillance. Prioritisation of urogenital schistosomiasis and female genital schistosomiasis surveillance is paramount and will generate valuable information that will guide the review and implementation of the current and old policies that were developed by the National Department of Health and stakeholders.

### Author summary

Urogenital schistosomiasis and female genital schistosomiasis continue to be among the neglected tropical diseases, posing challenges in accurately determining the true burden of these diseases. Female genital schistosomiasis, a gynaecological manifestation of *Schistosoma haematobium*, affects a significant number of women and young girls in regions where schistosomiasis is endemic. Some women suffering from female genital schistosomiasis are frequently misdiagnosed with conditions such as human papilloma virus (HPV) or other sexually transmitted infections (STIs). Integrating the surveillance of these neglected diseases into existing healthcare programmes, such as cervical cancer screening and HIV/STI and existing reproductive health services, offers opportunities for a comprehensive approach to patient screening and disease control. This study outlines the current neglected tropical diseases programme in South Africa and identifies opportunities for enhancing surveillance. Despite the neglect and challenges related to sustainability and under-reporting, the presence and utilization

of surveillance systems in South Africa highlight the country's potential for prioritization and integration to improve the prevention, surveillance, reporting, and management of schistosomiasis, as well as the introduction of surveillance for female genital schistosomiasis. Furthermore, establishing sentinel sites would provide opportunities to establish baseline prevalence and enhance monitoring of progress towards elimination.

## Introduction

Schistosomiasis is a neglected tropical disease that primarily affects impoverished communities in tropical Africa, the Middle East, Asia, Brazil, and South America that lack access to potable water and proper sanitation [1–10]. There is currently no vaccine for schistosomiasis, and current control strategies rely heavily on praziquantel treatment through mass drug administration for school-aged children [1,11,12]. Treatment with praziquantel is effective against the adult worm but has no effect on the presence of ova or migrating schistosomula to adults and the resultant patency [1,13–16]. Treatment alone will not keep the infection from recurring and, as a result, several rounds of treatment and discontinued exposure to risky water contact may be required for effectiveness [17]. Providing safe water to affected communities is critical to prevent re-infection following treatment. If left untreated, urogenital schistosomiasis can lead to complications which include the onset of female genital schistosomiasis (FGS) which presents in different morphologic sub-types (grainy sandy patches and homogenous yellow patches at 15 times magnification, abnormal blood vessels, and rubbery papules) [6,18,19].

It has been reported that sub-Saharan Africa accounts for only about 13% of the global population and that sub-Saharan Africa accounts for up to 90% of schistosomiasis cases and an estimated 280,000 deaths each year [1,11,12]. Schistosomiasis is endemic in the northern and eastern parts of South Africa, with *S. haematobium* being the most common species, and requires an active control programme [20]. Even though several countries have not achieved the WHO-recommended schistosomiasis preventive chemotherapy (PC) target of 75% of school-aged children, they have long implemented their programme in comparison to South Africa which is still struggling to implement the control pro-gramme and PC in the affected provinces and districts [1,12,21,22].

In the absence of an active control programme, the affected communities are vulnerable to complications such as FGS [18,19]. The last active control programme in South Africa was implemented in KwaZulu-Natal Province between 1997 and 2000 [23,24]. Understanding the country's schistosomiasis situation, followed by monitoring progress towards the World Health Organisation (WHO) specified elimination target of <1% proportion of heavy intensity infections is key to a successful control programme [11]. There is currently no programme data on infection intensity in South Africa, except as determined by some studies in the KwaZulu-Natal Province [25–28].

South Africa has still not accepted the WHO-donated medication for schistosomiasis because the drug is not registered in the country, and treatment is currently case-based [29,30]. Consequently, only those seeking medical attention would be treated, while others might dismiss the symptoms of genital discharge or dysuria as normal and live with these [31]. Currently, urine or stool specimens are collected from all suspected schistosomiasis cases seen in health facilities for laboratory confirmation [32]. However, laboratory data is not often published and is not being used to the greatest extent possible to inform policy, and those infected who do not seek medical attention will remain infected either silently or with symptoms they believe are normal.

The establishment of sentinel sites is one of the recommended common and best practices that provide opportuni-ties for improving surveillance systems [33]. If integrated with cervical cancer screening and other reproductive health services, these sentinel sites can facilitate the establishment of baseline prevalence of FGS through FGS screening and case management. Regular monitoring and evaluation surveys every 3 or 5 years are also appropriate for monitoring control progress [33]. In this paper, we explored secondary urogenital schistosomiasis data and conducted a review of all

available documents to identify surveillance best practices that can be institutionalized to improve and integrate schistosomiasis and FGS surveillance, prevention and control.

## Materials and methods

### Ethics statement

Formal consent was obtained in writing from the Biomedical Research Ethics Committee (BREC), University of KwaZulu-Natal (Ref BF029/07), KwaZulu-Natal Department of Health (Reference HRKM010–08) and the Regional Committee for Medical and Health Research Ethics (REC), South Eastern Norway (Ref 46907066a1.2007.535). Approval was granted from the National Department of Health to utilise data from the District Health Information System (DHIS) and from both the National Department of Health (NDoH) and the National Institute for Communicable Diseases to utilise data from the Notifiable Medical Conditions Surveillance System (NMCSS). Approval was granted from the National Health Laboratory Services (NHLS) to utilise data from the NHLS system. Data from the DHIS, NMCSS, and NHLS only included codes as a form of identification information.

### Study design, subjects and area

The study was conducted in South Africa, a democratic nation with three tiers of government: national, provincial, and local. The country is divided into nine provinces (Eastern Cape, Free State, Gauteng, KwaZulu Natal, Limpopo, Mpumalanga, North West, Northern Cape, and Western Cape), 52 districts, ten international airports, 54 land borders, and eight harbours.

### Descriptive study design

A descriptive study design was chosen to allow description of the current country's prevention and control strategies and to identify opportunities for improving surveillance and control of urogenital schistosomiasis and FGS. The study focused on analysis of secondary data from the NHLS for the period 2014–2018, secondary data from the NMCSS of the National Institute for Communicable Diseases (NICD) for the period 2017–2021, and secondary data from the DHIS of the NDOH for the period 2017–2021. The data is of all patients (all age groups) who visited healthcare facilities in all provinces of South Africa and who were suspected of having urogenital schistosomiasis based on urological symptoms. The flow of data surveillance is shown in Fig 1.

### Document review

All national documents on schistosomiasis surveillance and control were obtained from the National Department of Health. All the documents were reviewed through reading and extraction of surveillance recommended actions for schistosomiasis and FGS, to identify surveillance opportunities and best practices that can be institutionalised to improve disease control. The focus was on the documents from the National Department of Health only because they provide overall national guidance.

### Data sources, collection, processing, and analysis

There were three main sources of data, namely: the NHLS, the NMCSS, the DHIS, and previously published research data. The NHLS, NMCSS, and DHIS schistosomiasis data are reported from all nine provinces of South Africa; these three systems do not report on FGS. The NHLS and DHIS are electronic systems that collect data from patients who visit public health facilities, whereas the NMCSS is an electronic system that collects data from both public and private health facilities (including the NHLS data). As a result, the number of cases reported on the NMCSS is greater than the number of cases reported on the NHLS. The NMCSS is the primary reporting system for all notifiable medical conditions, including schistosomiasis, whereas the DHIS is a parallel system [34].

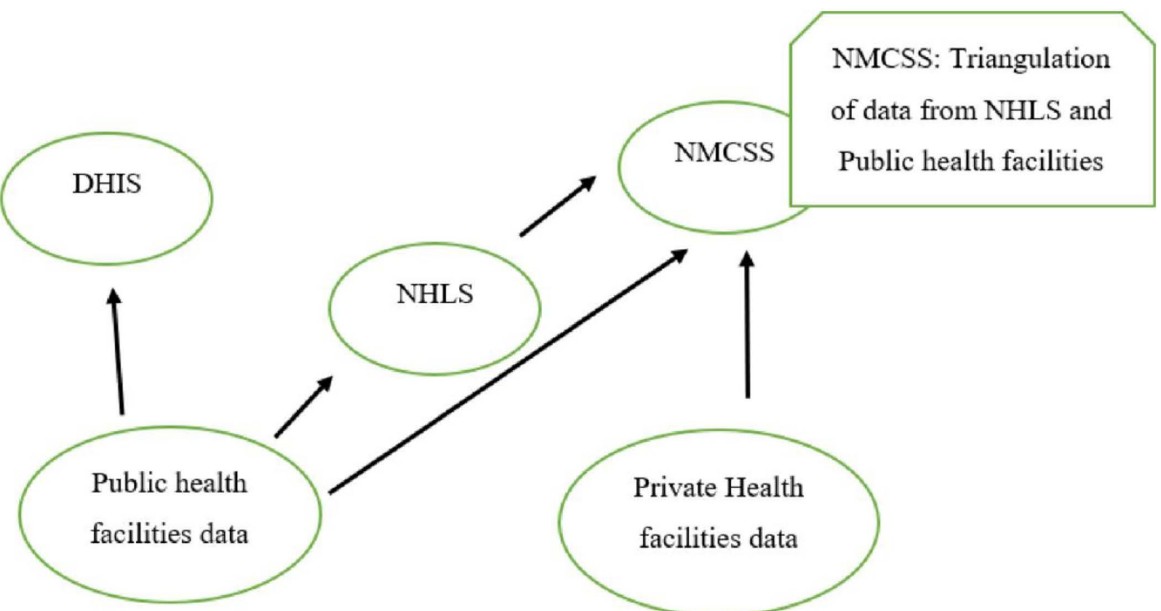

**Fig 1. Flow diagram for the schistosomiasis data reporting in South Africa.** DHIS - District Health Information System; NHLS - National Health Laboratory Services; NMCSS - Notifiable Medical Conditions Surveillance System.

Nation-wide laboratory data for patients tested for urogenital schistosomiasis and patients confirmed positive for urogenital schistosomiasis, for the period 2014 – 2018, was obtained from the NHLS. Nation-wide data for patients confirmed by laboratory to have urogenital schistosomiasis, for the period 2017 – 2021, was obtained from the NICD NMCSS. Nation-wide aggregated data for urogenital schistosomiasis cases reported to the DHIS, for the period 2017 – 2021, was obtained from the National Department of Health.

All data was extracted by the institutions that provided the data (the NHLS, NDoH, and NICD) and received in Excel Spreadsheets. The NHLS data were received with the following variables: episode number, ward name, facility name, facility type, facility classification, sub-district name, district name, province name, taken date, tested date, age tested in years, sex, specimen type, testing lab, erythrocytes, *Schistosoma haematobium* ova (observed denoted as OBS or OBSV, and not observed denoted as NOBS), urogenital schistosomiasis results. The NHLS data was cleaned by standardising variables and the records, was re-coded and the recordings "OBS, NOBS and OBSV" were standardised as "OBS" to indicate the presence of *Schistosoma* eggs and "NOBS" to indicate the absence of *Schistosoma* eggs.

The NMCSS data was received with the following variables: condition, case classification, Notifiable Medical Condition (NMC) Facility, Sub-District, District, Province, notifier, date symptoms, date diagnosis, notification date, folder number, patient first name, patient surname, gender, pregnancy status, date of birth, age in years, age category, citizenship, residential country, residential province, residential suburb, residential city town, travel history, travel1 country, travel1 province, travel1 city town, date travel1 from, date travel1 to, travel2 country, travel2 province, travel2 city town, date travel2 from, date travel2 to, symptom1, symptom2, treatment1, treatment2, vaccination status, date last vaccination, contact history, diagnosis method, specimen taken, specimen1 barcode, date specimen1 taken, patient status, date patient death, patient admission status, episode no. The NMCSS data was reviewed for errors and missing data using MS Excel filter and sorting, and duplicates using MS Excel Conditional Formatting. Patient name and surname were removed from the data.

The DHIS data was received with the following variables: province, district and year.

The data analysis was done using the Excel. The positivity rates and prevalence of urogenital schistosomiasis cases per year per district and province were determined using basic descriptive statistics (frequency and prevalence). The DHIS aggregated data was reported as received. The World Health Organization's endemicity measures were used to categorise districts, by using the calculated positivity rates from NHLS data, as low (<10%), moderate (≥10% but <50%), and high risk (≥50%).

We defined an active control programme as a programme that fully implements the control activities as recommended by the World Health Organization, namely: mass drug administration (prevention chemotherapy) to all communities at risk of infection including enrolled and unenrolled children of school-going age, adolescent girls and women of reproductive age, and all adults considered to be at risk living in endemic areas; water, sanitation and hygiene (WASH) activities; vector control; case-by-case management; surveillance; and other primary prevention measures to promote behavioural change, self-care and environmental management interventions [11]. The hypothesis was that integration of schistosomiasis and FGS into existing health programmes, such as STIs and reproductive health, would improve surveillance and understanding of the true burden of disease as well as its prevention and control. All the schistosomiasis and related documents were obtained from the National Department of Health Directorate: Communicable Diseases Control, as it is expected to coordinate the implementation of the schistosomiasis control programme [35]. These documents were reviewed to identify the existing policy and guidance in schistosomiasis and FGS surveillance and how they are being implemented.

Data from the NHLS, the NMCSS, and the DHIS were compared to previously published research data to see if there were any correlations in prevalence.

## Results

### Surveillance best practices and gaps

Although the country's schistosomiasis control programme is not active, efforts are made to develop guidelines and policies to guide implementation. Table 1 shows the best practices identified for schistosomiasis and FGS control in South Africa while Table 2 shows the gaps identified for schistosomiasis and FGS control in South Africa. The following surveillance activities were found to be recommended in the National Neglected Tropical Diseases (NTD) Master Plan, and they were directly quoted:

"Establish sentinel sites in selected communities and schools". "Introduce NTD surveillance data management within an integrated platform to ensure data flow is automated and available in real-time". "Develop surveillance guidelines and data management tools for NTDs". "Create cross border activities to strengthen surveillance between neighbouring countries and provinces". "Impact assessment surveys for Soil-transmitted Helminths and Schistosomiasis every 3 years and validation of data". "To carry out pharmacovigilance of NTD drugs including drug resistance surveillance". "Conduct baseline surveys". "Production and distribution of NTD Surveillance information, education and communication (IEC) materials in local languages". "Develop Surveillance IEC materials".

### Laboratory data and analysis

According to NHLS data, a total of 35,232 patients were suspected of having urogenital schistosomiasis between 2017 and 2018, with a urinary microscopy positivity rate of 10.4% (n = 3653). For the period 2014–2018, the positivity rate ranged from 0% to 36%, with Limpopo Province reporting the highest positivity rate, followed by Mpumalanga and KwaZulu-Natal (Table 3). In 2015, the North West Province reported a higher prevalence of 12% as compared to other reporting periods. Data analysis by district showed that although the number of patients tested in some districts was low, at least one district in each of the nine provinces reported a positivity rate of more than 10%. The majority of the districts

**Table 1. Best practices identified during document review, for schistosomiasis and FGS control in South Africa.**

| List of existing guiding documents: |
| --- |
| 1. Regulations relating to surveillance and control of notifiable medical conditions. These Regulations stipulate schistosomiasis as a category 2 notifiable medical condition that must be notified to the Department of Health by health care providers, public and private health laboratories within seven days of clinical or laboratory diagnosis [34]. The notification is done electronically through the NMCSS; facilities that do not have internet facilities have an opportunity to report using the paper-based format. |
| 2. South Africa National Master Plan for the Elimination of Neglected Tropical Diseases, 2019 – 2025. |
| 3. Standard Treatment Guidelines and Essential Medicines List for South Africa: Primary Health Care Level, 2020 Edition. |
| 4. Standard Treatment Guidelines and Essential Medicines List for South Africa Hospital Level, Adults 2019 Edition |
| 5. Standard Treatment Guidelines and Essential Medicines List for South Africa Paediatric Hospital Level, 2023 Edition |
| 6. Regular treatment of school-going children for soil-transmitted helminth infections and Bilharzia: Policy and implementation guidelines, 2008. |
| **Implementation of the surveillance** |
| 1. Data from NHLS, private health laboratories and private health care facilities are to be reported to the NMCSS. However, in addition to the NMCSS, the DHIS is a parallel system that is used to report routine schistosomiasis data on cases diagnosed and treated at public health facilities [36]. |
| 2. Data from the three systems revealed that some community members who suffer from urogenital schistosomiasis seek medical attention. The most common presenting symptom for urogenital schistosomiasis, according to NHLS and NMCSS data, is visible haematuria. |
| 3. Although the generic praziquantel is not yet available for mass drug administration for schistosomiasis control in South Africa, small quantities of praziquantel are available in case packaging at all levels of health care for case-based treatment and the treatment data is reported through the NHLS, NMCSS and DHIS [32,37]. |
| 4. As part of the surveillance strategies to determine the prevalence of schistosomiasis, the National Department of Health successfully mapped schistosomiasis in all provinces (KwaZulu-Natal and Limpopo provinces in 2016, Eastern Cape, Gauteng, Mpumalanga, and North West in 2018, Free State, Northern Cape and Western Cape provinces in 2019), and the findings were shared with stakeholders in all provinces except Northern Cape [38]. |
| 5. The NICD has a Centre for Emerging Zoonotic and Parasitic Diseases, with the Parasitology Reference Laboratory (PRL) as one of its divisions [39]. The PRL provides specialized parasitological diagnostic tests, such as those for schistosomiasis and this data is reported to the NMCSS. |

**Table 2. Gaps identified during document review, for schistosomiasis and FGS control in South Africa.**

| Gaps |
| --- |
| 1. FGS is not one of the notifiable medical conditions, and there is no FGS guideline and sentinel surveillance or any existing surveillance system. The only available data was from previously published research [5,40,41]. |
| 2. There was no sustainability in meetings and communication through the Neglected Tropical Diseases Task Team that was identified during document review [42]. As a result, the Neglected Tropical Diseases Master Plan and the Coordination Mechanism, which were developed in collaboration with stakeholders to guide the control program, were yet to be implemented and they were silent about FGS surveillance and control [35,43]. |
| 3. The unavailability of private health facility data from the NMCSS indicated that private health facilities are non-compliant with the Regulations relating to the surveillance and control of notifiable medical conditions (non-reporting). |
| 4. There is no integrated surveillance system. |
| 5. Schistosomiasis and FGS are neglected and there is a lack of implementation of some guiding documents; there is no supportive supervision and reporting is inconsistent across the three systems (NHLS, NMCSS, and DHIS), resulting in non-reporting or underreporting by most health facilities. This was confirmed by the lack of implementation of the WHO-recommended control activities, except case-by-case management and notification for surveillance. |
| 6. There was lack of a holistic and sustainable approach to FGS control, emphasizing the need for greater coordination between the NTD and sexual and reproductive health sectors [44–46]. |
| 7. Other gaps were identified during the analysis of the data from the three surveillance systems. These included data quality issues relating to standardizing reporting of variables such as clinical and laboratory diagnosis, and dates; and improving data quality by minimising data entry errors and ensuring complete data. |

had a positivity rate greater than 10% but less than 50% (moderate prevalence). The districts that had a positivity rate of ≥50% (high risk) included Amathole District in 2015 and 2016 in the Eastern Cape Province, Mopani District in 2017 in Limpopo Province, OR Tambo District in 2014 in Eastern Cape Province, and Vhembe District in 2017 in Limpopo

**Table 3. Clinically diagnosed urogenital schistosomiasis in 9 provinces and urinary microscopy positivity rate, reported through the National Health Laboratory Services, South Africa, 2014–2018.**

| Prov | 2014 | | | 2015 | | | 2016 | | | 2017 | | | 2018 | | |
|------|------|------|--------|------|------|--------|------|------|--------|------|------|--------|------|------|--------|
| | NOBS[a] | OBS[b] | P/rate[c] | NOBS[a] | OBS[b] | P/rate[c] | NOBS[a] | OBS[b] | P/rate[c] | NOBS[a] | OBS[b] | P/rate[c] | NOBS[a] | OBS[b] | P/rate[c] |
| EC | 550 | 47 | 9% | 728 | 61 | 8% | 842 | 72 | 9% | 884 | 77 | 9% | 1023 | 96 | 9% |
| FS | 25 | 0 | 0% | 32 | 1 | 3% | 36 | 0 | 0% | 33 | 0 | 0% | 28 | | 0% |
| GP | 1249 | 24 | 2% | 1322 | 35 | 3% | 1129 | 35 | 3% | 1163 | 28 | 2% | 1009 | 36 | 4% |
| KZN | 1928 | 284 | 15% | 2689 | 293 | 11% | 2658 | 259 | 10% | 2106 | 214 | 10% | 2037 | 174 | 9% |
| LP | 629 | 160 | 25% | 545 | 139 | 26% | 563 | 197 | 35% | 458 | 166 | 36% | 413 | 124 | 30% |
| MP | 1262 | 202 | 16% | 1218 | 223 | 18% | 1037 | 206 | 20% | 904 | 203 | 22% | 880 | 188 | 21% |
| NW | 108 | 6 | 6% | 95 | 11 | 12% | 77 | 6 | 8% | 88 | 5 | 6% | 94 | 2 | 2% |
| NC | 12 | 1 | 8% | 24 | | 0% | 12 | 1 | 8% | 10 | 1 | 10% | 12 | | 0% |
| WC | | | | 156 | 10 | 6% | 473 | 22 | 5% | 531 | 18 | 3% | 507 | 26 | 5% |
| Total | **5763** | **724** | | **6809** | **773** | | **6827** | **798** | | **6177** | **712** | | **6003** | **646** | |

[a]Presented with symptoms of urogenital schistosomiasis but Schistosomiasis eggs not observed,

[b]Presented with symptoms of urogenital schistosomiasis and Schistosomiasis confirmed,

[c]Positivity rate, NOB – schistosome eggs not observed, OBS - schistosome eggs observed, P/rate – Positivity rate, blank spaces: no data.

Province. West Coast District also reported a positivity rate of more than 50% in 2015, but only two patients were tested. The positivity rate dropped from 12.6% in 2017 to 10.8 in 2018; however, due to inconsistent reporting, it was difficult to determine whether this was a true decrease.

Data from the NMCSS system demonstrate that urogenital schistosomiasis cases are reported every year in South Africa, in both public and private health facilities (Fig 2). Of the total of 56,529 cases reported, the majority were laboratory-confirmed by microscopy, but 20 cases were diagnosed by signs and/or symptoms only (suspected cases), and 12 cases had no diagnosis. The total number of cases reported through the NMCSS in the country increased during 2017–2019 (N = 8823 in 2017, N = 14960 in 2018, N = 15032 in 2019), and decreased during the COVID-19 pandemic

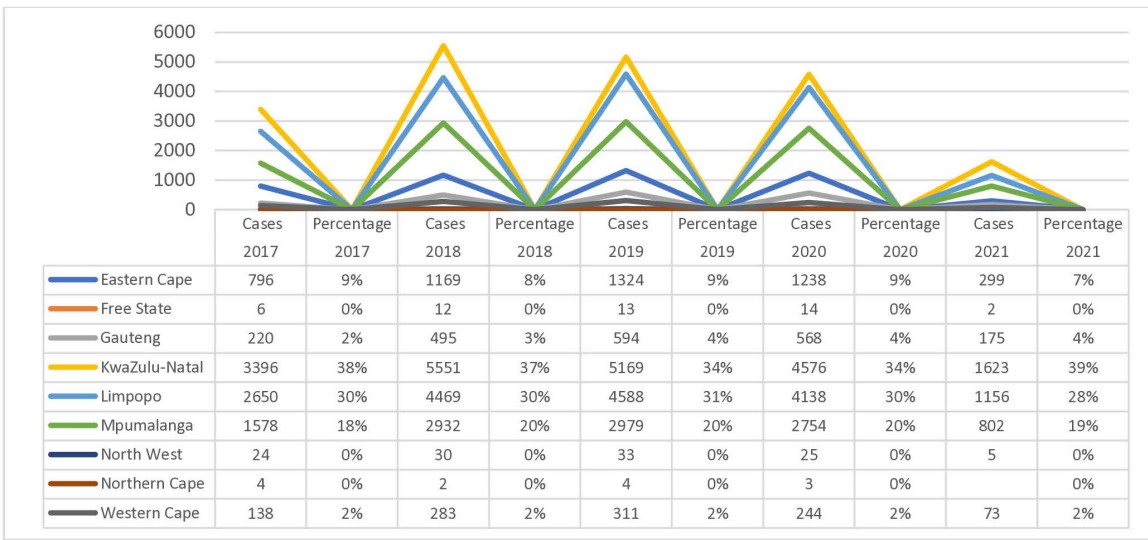

**Fig 2. Provincial urogenital schistosomiasis per year, reported through NMCSS, South Africa, 2017–2021.**

(N = 13574 in 2020, and N = 4140 in 2021). Over the five years, 2017–2021, most of the cases were reported from KwaZulu-Natal Province (N = 20315 (36%) and over 5,000 cases reported each year in 2018 and 2019), followed by Limpopo Province (N = 17001, 30%), Mpumalanga (N = 11045, 20%) and Eastern Cape (N = 4826, 9%). None of these were investigated for FGS.

**Blank spaces: No data**

According to the NMCSS data, all districts in the three most affected provinces (KwaZulu-Natal, Limpopo and Mpumalanga) reported cases of urogenital schistosomiasis confirmed by microscopy. Ehlanzeni District in Mpumalanga Province reported the highest cases as compared to other districts (N = 1462 in 2017, N = 2735 in 2018, N = 2780 in 2019, N = 2473 in 2020 and N = 714 in 2021), followed by Vhembe District in Limpopo Province (Fig 3).

For the period 2017 – 2021, more males (N = 40172; 71%) tested positive (laboratory confirmation by microscopy) for urogenital schistosomiasis as compared to females (N = 14456; 26%) (Fig 4 for annual data).

Most cases were reported among the age group 10–14 years (>30% of the positive cases were found in this age group, each year during 2017–2021), followed by the age group 15–19 years (>19%) (Fig 5). Other age groups that were frequently affected were 5–9 years (between 12% and 16% of the positive cases were found in this age group each year), and 20–24 (between 9% and 11%).

As shown in Fig 6, since 2017, the DHIS surveillance system has reported fluctuating numbers of laboratory confirmed urogenital schistosomiasis cases. Similar to the data reported through the NMCSS, over the five years (2017–2021), KwaZulu-Natal contributed the most cases followed by Limpopo and Mpumalanga (Western Cape did not report any cases).

The analysis of DHIS data by district showed that within the known endemic provinces, Vhembe district in Limpopo reported the majority of cases for the period 2017–2021, followed by Mopani district in Limpopo and Ehlanzeni district in Mpumalanga.

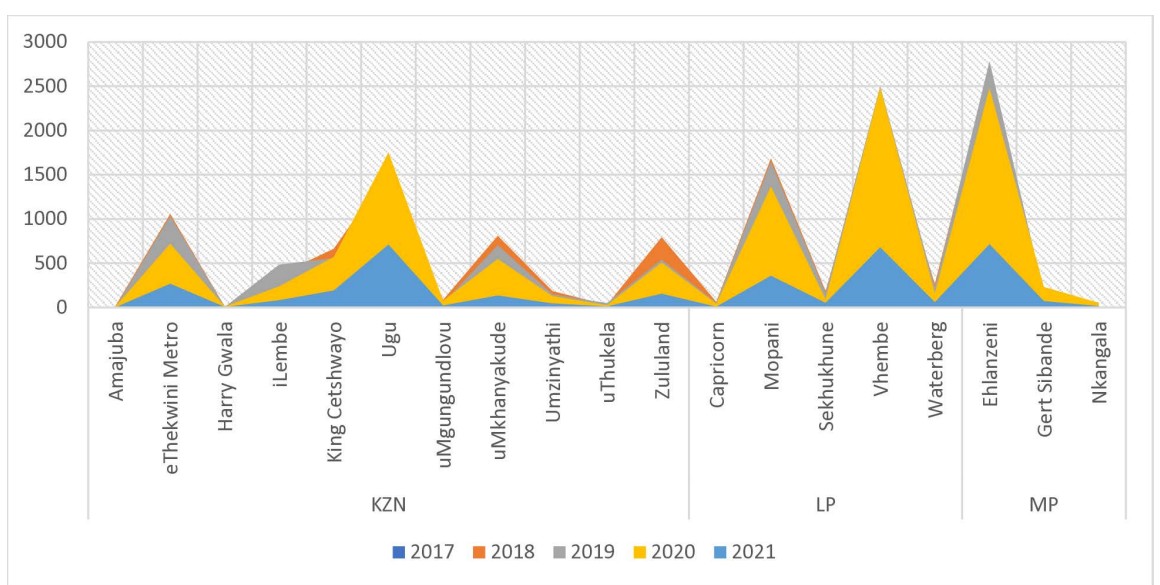

**Fig 3. District trends of urogenital schistosomiasis per year in the most affected provinces, reported through NMCSS, South Africa, 2017–2021.**

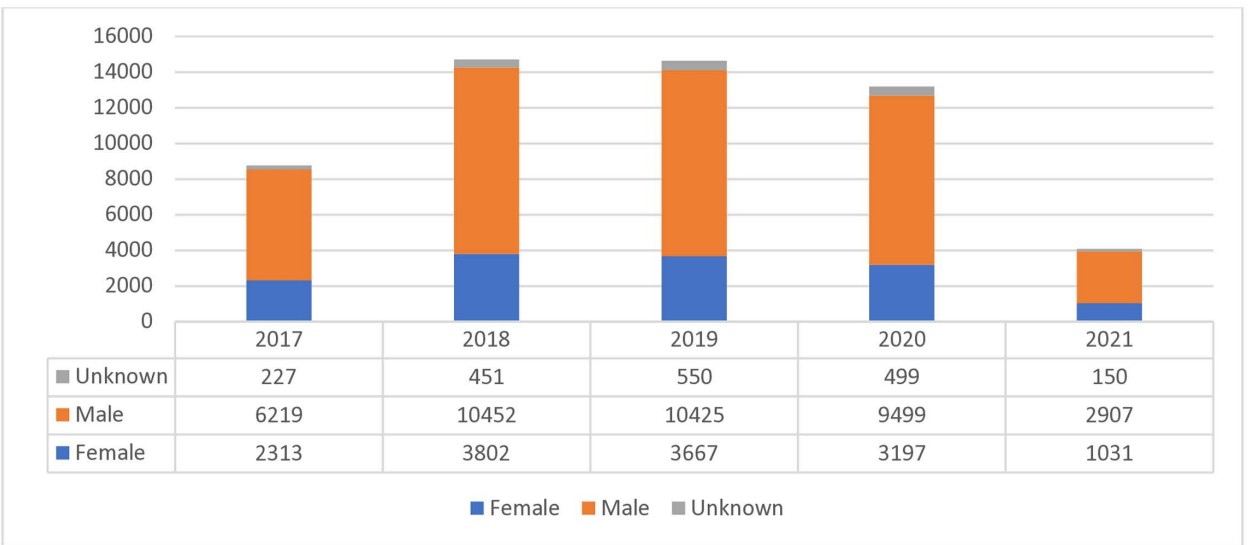

**Fig 4. Annual urogenital schistosomiasis cases by gender, reported through NMCSS, South Africa, 2017–2021.**

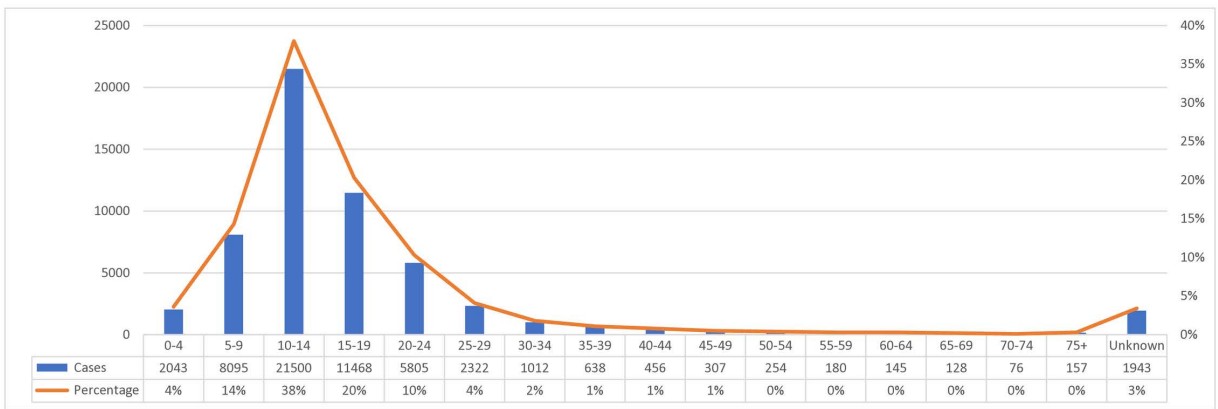

**Fig 5. Urogenital schistosomiasis positive by age category, reported through NMCSS, South Africa, 2017–2021.**

## Discussion and conclusion

The findings suggest that South Africa has surveillance best practices that offer opportunities for the country to improve urogenital schistosomiasis and FGS prevention and management. The analysis of data from this study indicates that despite a significant underreporting challenge, the inclusion of schistosomiasis as a notifiable medical condition, along with the presence of electronic reporting systems and a diagnostic laboratory network, provides an opportunity for improving urogenital and FGS surveillance in South Africa.

To understand the true burden of urogenital schistosomiasis and FGS, it is important to establish sentinel sites to establish the baseline prevalence and track progress toward elimination. Sentinel sites should be established in all local municipalities where schistosomiasis is prevalent. At these sites, schistosomiasis and FGS screening, and case management should be integrated into cervical cancer screening and HIV/STI and existing reproductive health services. Additionally, regular monitoring and evaluation surveys every 3–5 years are recommended to track control progress [33]. China's

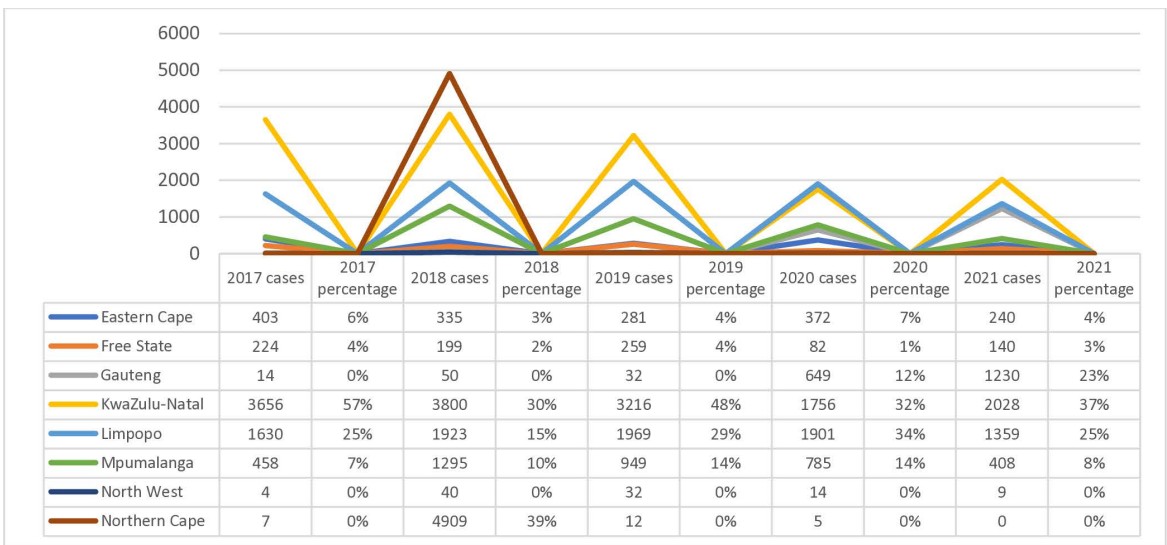

**Fig 6. Urogenital Schistosomiasis cases by provinces, reported through DHIS, South Africa, 2017–2021.**

| | 2017 cases | 2017 percentage | 2018 cases | 2018 percentage | 2019 cases | 2019 percentage | 2020 cases | 2020 percentage | 2021 cases | 2021 percentage |
|---|---|---|---|---|---|---|---|---|---|---|
| Eastern Cape | 403 | 6% | 335 | 3% | 281 | 4% | 372 | 7% | 240 | 4% |
| Free State | 224 | 4% | 199 | 2% | 259 | 4% | 82 | 1% | 140 | 3% |
| Gauteng | 14 | 0% | 50 | 0% | 32 | 0% | 649 | 12% | 1230 | 23% |
| KwaZulu-Natal | 3656 | 57% | 3800 | 30% | 3216 | 48% | 1756 | 32% | 2028 | 37% |
| Limpopo | 1630 | 25% | 1923 | 15% | 1969 | 29% | 1901 | 34% | 1359 | 25% |
| Mpumalanga | 458 | 7% | 1295 | 10% | 949 | 14% | 785 | 14% | 408 | 8% |
| North West | 4 | 0% | 40 | 0% | 32 | 0% | 14 | 0% | 9 | 0% |
| Northern Cape | 7 | 0% | 4909 | 39% | 12 | 0% | 5 | 0% | 0 | 0% |

experience with schistosomiasis offers valuable lessons for South Africa. After achieving significant elimination milestones since 1985, China launched a national surveillance project in the 1990s and successfully eliminated the disease between 1985 and 1995 [47–49]. In 2020, the China updated its National Surveillance Plan to address the current low endemicity and identify areas at risk of transmission [47–49].

Although the NMCSS and the NHLS systems are integrated, the DHIS was found to be independent of these two surveillance systems. The NHLS and DHIS are accessible by all public health facilities whereas the NMCSS is accessible by both public and private healthcare facilities and laboratories. There is therefore a need for an integrated surveillance system to maximize output and efficiency. The existence of Regulations relating to notifiable medical conditions provides another opportunity for improvement in reporting and reporting standards (including data quality improvements). This is because all public and private laboratories are expected to comply with these Regulations and notify all priority medical conditions including schistosomiasis. However, FGS still needs to be indicated as one of the priority medical conditions to facilitate the establishment of its surveillance, and to facilitate public and private laboratory and healthcare facility compliance in reporting diagnosis [34].

Data from the three systems is similar to the data reported by previous research and the mapping exercise conducted by NDoH in terms of the provinces that report the most cases and the most affected age groups [25–28,50–52]. The reported data are also very similar to data reported by other countries, indicating that the data may be reasonably reliable despite the selection. Generally, data analysis of this study suggests that KwaZulu-Natal, Limpopo and Mpumalanga provinces are the most affected provinces in South Africa. The National Department of Health's mapping of schistosomiasis in all nine provinces of South Africa demonstrates that the country has some capacity to implement the urogenital schistosomiasis and FGS control program. As a follow-up of the mapping findings, the WHO treatment strategy and other prevention and control measures need to be implemented in all at-risk communities. The country needs to prioritise implementation of the Neglected Tropical Diseases Master Plan and the Coordination Mechanism and develop a monitoring and evaluation plan. The WHO FGS pocket atlas for clinical healthcare professionals is a good resource for training clinicians on FGS screening and management [19].

South Africa has a sizable population (27%) that relies on the private health sector for medical care [53], and according to the NMCSS, these communities are also prone to schistosomiasis infection. As a result, capacity building and

enforcement of the Regulations relating to surveillance and control of notifiable medical conditions would ensure that public and private health sectors comply with reporting. This will assist the country in better understanding the disease burden to inform control measures.

The Laboratory capacity also provides opportunities for operationalising the sentinel sites proposed by Nemungadi et al. [33].The sentinel sites should be capacitated through clinician and other health worker training, as well as the provision of equipment and tools for screening for schistosomiasis and FGS, treatment of at-risk community members, data management, and reporting. The mapping exercise data and the previously published schistosomiasis and FGS research data support the need for establishing sentinel sites to better understand disease burden [25–28,50–52]. Operationalisation of these sentinel sites may assist the country in establishing the FGS baseline prevalence, confirming the urogenital schistosomiasis prevalence data from published research and from the mapping exercise conducted between 2016 and 2019 [33]. The overall program will enable the country to implement a monitoring and evaluation program and track progress toward WHO schistosomiasis elimination targets. The monitoring and evaluation tool requires monitoring indicators related to baseline urogenital schistosomiasis and FGS prevalence, quarterly and annual case detection rate for urogenital schistosomiasis and FGS, cases of local origin and those that are imported, the population at risk, and treatment coverage. For the sentinel sites to be successful, dedicated staff and supportive supervisory visits will be key.

The availability of praziquantel for case-based treatment in all health facilities demonstrates some country's capacity to manage schistosomiasis cases. However, case-based treatment for schistosomiasis has its own set of challenges due to low treatment demand and poor health-seeking behaviour among most affected people [31]. As a result, a large number of community members remain untreated and probably have FGS complications and continue to shed eggs and potentially contaminate water bodies. Annual testing and the positivity rate are very low when compared with the population size and research conducted in the past in KwaZulu-Natal, Limpopo and Mpumalanga provinces. The age tested also reveals that most young females (below 16 years) may be missing the schistosomiasis and FGS treatment benefits. Consequently, this missed age group are at high risk of FGS due to lack of mass drug administration and a FGS control programme, and the social difficulties of these young females in accessing gynaecological investigations. In this context, sentinel sites may be a better option to target hot spots and conduct targeted mass drug administration to prevent schistosomiasis complications from FGS; this option may be cheaper than a blanket mass drug administration using the currently registered praziquantel. At the sentinel sites, urogenital schistosomiasis testing, and FGS screening could be improved by routinely selecting and screening community members who visit healthcare facilities in affected or at-risk areas, based on a history of exposure. As a result, the demand for treatment will increase and it would put the country at an advantage in demonstrating the cost-effectiveness of mass drug administration and motivating the availability of WHO-donated medication.

The lack of sustainability in meetings and communication through the Neglected Tropical Diseases Task Team and the lack of FGS as one of the main agenda items necessitates the need for awareness about FGS among stakeholders including programme managers, clinicians and community members. A successful schistosomiasis control program demands multi-stakeholder and multi-sectoral collaboration. The existence of a control programme would address issues related to underreporting. Reporting issues that were noted could be addressed by the existence of a sustainable control programme. These reporting issues included the decline in case reporting during the COVID-19 pandemic, non-reporting by some provinces (possibly due to lack of zero reporting), and the gender disparity in urogenital schistosomiasis prevalence. The interest of some academic institutions in schistosomiasis research, such as the University of KwaZulu-Natal and the University of Venda, provides an opportunity for the country to involve stakeholders in reporting, managing and controlling the disease, as well as FGS as a complication. The NICD Centre for Emerging Zoonotic and Parasitic Diseases, as well as other government agencies like Water and Sanitation and Local Municipalities, would provide excellent opportunities for improving operational research and public health. Academic and research institutions bring research skills that can help with program monitoring and evaluation.

Schistosomiasis has been linked to increased HIV transmission, and some FGS cases have been misdiagnosed as STIs [4,13]. As a result, to maximize effectiveness and efficiency, the overall schistosomiasis control program could be integrated with the HIV, STI, and reproductive health programs. As recommended by Nemungadi *et al* in 2022, integrating schistosomiasis into these existing health programmes would result in improved access to schistosomiasis prevention and treatment services, as well as decreased risk of new FGS and contribute to enhancing population health in the affected communities [33]. WHO recommends the use of existing platforms such as schools and antenatal clinics, informal training centres, adolescent-friendly clinics and services and educational institutions for integration and to reach adolescent girls and women of reproductive age [54].

Limitations to the findings include the fact that only data from the existing surveillance system and a review of the existing documents were considered instead of involving the qualitative component of interviewing the public and private health implementers at all levels of the health system (national, province, districts, sub-districts, and health facilities). The analysis of the cases reported may not reflect a meaningful and true reflection of the true burden of disease because schistosomiasis and FGS are neglected and there could be a serious challenge of underreporting. Furthermore, it was not clear from the DHIS if the data was for patients who were clinically or laboratory diagnosed or a combination of both. There was no triangulation of data. The study focused on descriptive analysis and stakeholder interviews were not conducted to gain further insight into the process. Further investigations are needed either through focus group discussions and/or individual interviews or both with implementers at all levels of the health system to understand factors influencing compliance with the Regulations relating to notifiable medical conditions.

## Supporting information

**S1 Data.** **This is the data set from the NMCSS.**
(XLSX)

**S2 Data.** **This is the data set from the DHIS.**
(XLSX)

**S3 Data.** **This is the data set from the NHLS.**
(XLSX)

## Acknowledgments

We thank the National Department of Health, NHLS, NICD for providing the urogenital schistosomiasis data. We are appreciative of the support with health facility location data from Silindile Gagai and other staff at BRIGHT Research in KwaZulu-Natal, South Africa.

## Author contributions

**Conceptualization:** Takalani Girly Nemungadi, Eyrun F Kjetland.

**Formal analysis:** Takalani Girly Nemungadi.

**Methodology:** Takalani Girly Nemungadi.

**Project administration:** Takalani Girly Nemungadi.

**Supervision:** Saloshni Naidoo, Eyrun F Kjetland.

**Validation:** Takalani Girly Nemungadi.

**Visualization:** Takalani Girly Nemungadi.

**Writing – original draft:** Takalani Girly Nemungadi.

**Writing – review & editing:** Takalani Girly Nemungadi, Tsakani Furumele, Absalom Mwazha, Myra Taylor, Saloshni Naidoo, Eyrun F Kjetland.

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
