## [Decision Letter · Decision Letter 0]

PNTD-D-24-01521INSTITUTIONALISING SCHISTOSOMIASIS SURVEILLANCE: BEST PRACTICES TO IMPROVE FEMALE GENITAL AND URINARY SCHISTOSOMIASIS CONTROL IN SOUTH AFRICAPLOS Neglected Tropical Diseases Dear Dr. Nemungadi, Thank you for submitting your manuscript to PLOS Neglected Tropical Diseases. After careful consideration, we feel that it has merit but does not fully meet PLOS Neglected Tropical Diseases's publication criteria as it currently stands. Therefore, we invite you to submit a revised version of the manuscript that addresses the points raised during the review process. Please submit your revised manuscript within 30 days May 25 2025 11:59PM. If you will need more time than this to complete your revisions, please reply to this message or contact the journal office at plosntds@plos.org. Please include the following items when submitting your revised manuscript:

* A rebuttal letter that responds to each point raised by the editor and reviewer(s). You should upload this letter as a separate file labeled 'Response to Reviewers '. This file does not need to include responses to any formatting updates and technical items listed in the 'Journal Requirements' section below.

 * A marked-up copy of your manuscript that highlights changes made to the original version. You should upload this as a separate file labeled 'Revised Manuscript with Track Changes '. * An unmarked version of your revised paper without tracked changes. You should upload this as a separate file labeled 'Manuscript '. If you would like to make changes to your financial disclosure, competing interests statement, or data availability statement, please make these updates within the submission form at the time of resubmission. Guidelines for resubmitting your figure files are available below the reviewer comments at the end of this letter. We look forward to receiving your revised manuscript. Kind regards, Gabriel Rinaldi, M.D., Ph.D.Section EditorPLOS Neglected Tropical Diseases Eva ClarkSection EditorPLOS Neglected Tropical Diseases

Shaden Kamhawi

co-Editor-in-Chief

Paul Brindley

co-Editor-in-Chief

**Journal Requirements:**

At this stage, the following Authors/Authors require contributions: Takalani Girly Nemungadi. Please ensure that the full contributions of each author are acknowledged in the "Add/Edit/Remove Authors" section of our submission form.

**Reviewers' comments:** Reviewer's Responses to Questions

**Key Review Criteria Required for Acceptance?**

**Methods:**

-Are the objectives of the study clearly articulated with a clear testable hypothesis stated?

-Is the study design appropriate to address the stated objectives?

-Is the population clearly described and appropriate for the hypothesis being tested?

-Is the sample size sufficient to ensure adequate power to address the hypothesis being tested?

-Were correct statistical analysis used to support conclusions?

-Are there concerns about ethical or regulatory requirements being met?

Reviewer #1: **Minor Revision**

The objectives of the study are clearly articulated, and the study design is appropriate for addressing these objectives, providing a solid framework for investigation. There are no concerns regarding ethical or regulatory requirements. However, please consider the following suggestions:

1. **Line 195**: The rationale behind the study would benefit from further elaboration on the scope of the control measures considered, particularly regarding deworming efforts for adolescent girls and women of reproductive age (see WHO Policy Brief). What was the reason for omitting this aspect from your study’s focus?

2. **Address as limitations**:

a. **Data Analysis**: There was no triangulation of data. Stakeholder interviews were not conducted to gain further insight into the process, but this is addressed in the limitation chapter.

b. The authors have utilised descriptive analysis; please mention this as an limitation into the limitations chapter of your study.

Reviewer #2: Objectives and Hypothesis:

The manuscript outlines the objectives but lacks a clear, testable hypothesis.

Suggestion: Explicitly state a hypothesis related to the integration of FGS into existing surveillance systems and expected outcomes.

Study Design Appropriateness:

The descriptive study design is appropriate for analyzing surveillance data, but there is no control or comparison group.

Suggestion: Justify the choice of a descriptive study design and discuss how it limits or strengthens the conclusions.

Population Description:

The study examines surveillance data from public health facilities but does not fully describe the demographics of the affected population.

Suggestion: Provide more details on the geographic, demographic, and socioeconomic characteristics of the population covered by the surveillance systems.

Sample Size and Power Analysis:

The manuscript does not discuss sample size calculations or statistical power.

Suggestion: Justify whether the number of cases reported is sufficient for meaningful analysis, and discuss potential biases due to underreporting.

Statistical Analysis:

The manuscript mainly relies on descriptive statistics without more advanced statistical testing.

Suggestion: If feasible, incorporate statistical comparisons (e.g., chi-square tests for differences in reporting rates between provinces or health sectors) to strengthen conclusions.

Data Sources and Collection: Ensure clarity on how data were extracted, cleaned, and analyzed, including potential limitations in data accuracy.

Ethical Considerations: While ethical approvals are noted, discuss how patient confidentiality and data protection were ensured.

Potential Biases: Address limitations such as reporting inconsistencies across surveillance systems and how they might affect findings.

**Results** :

-Does the analysis presented match the analysis plan?

-Are the results clearly and completely presented?

-Are the figures (Tables, Images) of sufficient quality for clarity?

Reviewer #1: **Minor Revision**

The analysis presented aligns with the analysis plan, and the results are clearly and comprehensively displayed.

1. **Revision of Table 2**:

I recommend adding a holistic and sustainable approach to FGS control to the list of gaps, emphasizing the need for greater coordination between the NTD and SRH sectors. Several articles can support this claim. Below are a few for your consideration.

•Engels D, Hotez PJ, Ducker C, Gyapong M, Bustinduy AL, Secor WE, Harrison W, Theobald S, Thomson R, Gamba V, Masong MC. Integration of prevention and control measures for female genital schistosomiasis, HIV and cervical cancer. Bulletin of the World Health Organization. 2020 Sep 9;98(9):615.

•Williams CR, Seunik M, Meier BM. Human rights as a framework for eliminating female genital schistosomiasis. PLOS Neglected Tropical Diseases. 2022 Mar 3;16(3):e0010165.

•Pillay LN, Umbelino-Walker I, Schlosser D, Kalume C, Karuga R. Minimum service package for the integration of Female Genital Schistosomiasis into sexual and reproductive health and rights interventions. Frontiers in Tropical Diseases. 2024 Jun 17;5:1321069.

2.**Improve Figures 4 and 5**.

Currently, both absolute values (number of cases) and proportions (percentages) are presented on the same axis, making it difficult to understand. Consider using separate y-axes for the absolute numbers and the percentages, or improve the labelling of the axes for better clarity.

Reviewer #2: The results align with the stated objectives and methodology but could benefit from more explicit connections to the research questions.

Suggestion: Clearly reference the analysis plan in the results section to demonstrate how findings support the study's aims.

The results are well-structured and include key findings from multiple data sources. However, some sections could provide additional interpretation of the trends observed.

Suggestion: Expand on the implications of the data trends, particularly the declining case reports during the COVID-19 pandemic and the gender disparity in schistosomiasis prevalence.

The figures and tables are informative, but some require clearer labeling and improved consistency in formatting.

Suggestion: Ensure uniformity in table structures and provide concise yet informative figure captions.

**Conclusions:**

-Are the conclusions supported by the data presented?

-Are the limitations of analysis clearly described?

-Do the authors discuss how these data can be helpful to advance our understanding of the topic under study?

-Is public health relevance addressed?

Reviewer #1: **Major Revision**

Your discussion provides a solid foundation for improving the management of schistosomiasis and FGS in South Africa. Please consider the following suggestions:

1. I strongly recommend adding a paragraph that discusses the potential benefits of integrating schistosomiasis and FGS surveillance into existing sexual and reproductive health and rights (SRHR) programs. This addition would present a more comprehensive approach to addressing both schistosomiasis and FGS in South Africa and beyond.

2. Include a discussion on aligning national policies with international frameworks regarding SRHR and NTDs.

3. Consider including a comparison with other countries that have institutionalized urogenital schistosomiasis surveillance, highlighting what lessons can be learned from their experiences.

4. Address the limitations of the descriptive analysis in your discussion.

5. Add a conclusion that summarizes specific recommendations for policymakers, international organizations, and local stakeholders to support the institutionalization of urogenital schistosomiasis surveillance.

Reviewer #2: The conclusions are generally aligned with the findings but could be strengthened by explicitly linking key results to recommendations.

Suggestion: Clearly restate the most significant findings and how they support the proposed recommendations.

The manuscript acknowledges some limitations, such as inconsistent reporting, but does not fully explore their impact on the findings.

Suggestion: Expand on how limitations may affect the interpretation of results and suggest ways to address them in future research.

The discussion highlights the need for improved FGS surveillance, but it lacks a detailed explanation of how the findings contribute to broader knowledge.

Suggestion: Provide a clearer discussion on how this study builds on previous research and offers new insights into schistosomiasis control.

The manuscript emphasizes the importance of integrating FGS surveillance into existing health systems, which is a critical public health issue.

Suggestion: Strengthen the discussion on the policy implications of these findings, particularly in the context of neglected tropical disease control strategies. Authors should elaborate on practical steps for integrating FGS surveillance into existing systems.

**Editorial and Data Presentation Modifications?**

Reviewer #1: **Minor Revision**

The document uses various fonts and font sizes. Please standardise them to ensure uniformity throughout.

Reviewer #2: Abstract: Consider briefly mentioning that FGS is currently not a notifiable medical condition and how this impacts surveillance.

Figures & Tables: Ensure consistency in the way data are reported across figures and tables for clarity.

Grammar & Style: Some sentences are complex and lengthy; consider breaking them into shorter, clearer statements for better readability.

The writing is dense and technical, making it harder for non-specialists to follow. Some sections (e.g., Discussion) are repetitive, reiterating findings without additional interpretation.

Suggestion: Improve conciseness and readability, especially in the Discussion. Use bullet points or summarizing statements where appropriate.

**Summary and General Comments:**

Reviewer #1: **Minor Revision**

The paper is well-organized, with a clear and logical structure that effectively guides the reader through the research process. The writing is coherent and concise, making complex ideas accessible without sacrificing depth or detail. The paper highlights critical gaps in surveillance, treatment, and intersectoral collaboration, and make a strong case for improving the existing systems. Minor revisions are needed. Please consider expanding through the paper on the need for integration surveillance into SRHR programs.

*Introduction*

1.The introduction is comprehensive and provides a solid foundation for the study by addressing the burden, challenges, and gaps in schistosomiasis control in South Africa. However, it would be further enhanced by adding a paragraph discussing the potential for integrating schistosomiasis surveillance into Sexual and Reproductive Health (SRH) programs. Additionally, a brief reference to the recently published WHO policy brief titled "Deworming adolescent girls and women of reproductive age. Policy brief" could strengthen the rationale for such integration and highlight its relevance to addressing FGS and advancing public health strategies.

2.Please consider changing the term “urinary schistosomiasis” and “schistosomiasis” to "urogenital schistosomiasis”. It is more appropriate since your manuscript works with the intersection of FGS, an infection in both the urinary and genital organs. I suggest also updating the legend and caption in figures and tables.

Reviewer #2: This manuscript presents a valuable analysis of existing schistosomiasis surveillance systems in South Africa, highlighting the lack of institutionalized surveillance for female genital schistosomiasis (FGS). The study is well-structured and informative, with clear objectives and a thorough review of available data. However, there are areas that require further clarification and refinement to enhance its impact and clarity. Addressing the points above will improve its clarity, impact, and practical relevance for policymakers and researchers.

We appreciate the authors' valuable work and encourage revisions to strengthen the manuscript further.

PLOS authors have the option to publish the peer review history of their article (what does this mean? ). If published, this will include your full peer review and any attached files.

**Do you want your identity to be public for this peer review?** For information about this choice, including consent withdrawal, please see our Privacy Policy .

Reviewer #1: No

Reviewer #2: No

---

## [Editor Report · Decision Letter 1]

Dear Ms Nemungadi,

We are pleased to inform you that your manuscript 'INSTITUTIONALISING UROGENITAL SCHISTOSOMIASIS SURVEILLANCE: BEST PRACTICES TO IMPROVE FEMALE GENITAL AND URINARY SCHISTOSOMIASIS CONTROL IN SOUTH AFRICA' has been provisionally accepted for publication in PLOS Neglected Tropical Diseases.

Best regards,

Francesca Tamarozzi

Section Editor

Gabriel Rinaldi

Section Editor

Shaden Kamhawi

co-Editor-in-Chief

Paul Brindley

co-Editor-in-Chief

---

## [Editor Report · Acceptance letter]

Dear Ms Nemungadi,

We are delighted to inform you that your manuscript, "INSTITUTIONALISING UROGENITAL SCHISTOSOMIASIS SURVEILLANCE: BEST PRACTICES TO IMPROVE FEMALE GENITAL AND URINARY SCHISTOSOMIASIS CONTROL IN SOUTH AFRICA," has been formally accepted for publication in PLOS Neglected Tropical Diseases.

Best regards,

Shaden Kamhawi

co-Editor-in-Chief

Paul Brindley

co-Editor-in-Chief
